# Knowledge-dependent optimal Gaussian strategies for phase estimation

Ricard Ravell Rodríguez[1] and Simon Morelli[2]

**1** Institute for Cross-Disciplinary Physics and Complex Systems IFISC (UIB-CSIC), Campus Universitat Illes Balears, E-07122 Palma de Mallorca, Spain

**2** Atominstitut, Technische Universität Wien, Stadionallee 2, 1020 Vienna, Austria

## Abstract

When estimating an unknown phase rotation of a continuous-variable system with homodyne detection, the optimal probe state strongly depends on the value of the estimated parameter. In this article, we identify the optimal pure single-mode Gaussian probe states depending on the knowledge of the estimated phase parameter before the measurement. We find that for a large prior uncertainty, the optimal probe states are close to coherent states, a result in line with findings from noisy parameter estimation. However, with increasingly precise estimates of the parameter, it becomes beneficial to put more of the available energy into the squeezing of the probe state. Surprisingly, there is a clear jump, where the optimal probe state changes abruptly to a squeezed vacuum state, which maximizes the Fisher information for this estimation task. We use our results to study repeated measurements and compare different methods to adapt the probe state based on the changing knowledge of the parameter according to the previous findings.

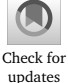
# 1   Introduction

Interferometry, i.e., the estimation of the relative phase between two modes of light, is a well-known problem in metrology [1]. In quantum theory, the problem is reformulated as the estimation of the phase of the quantum state of an electromagnetic field [2, 3] and is still a paradigmatic task in quantum metrology [4, 5]. To estimate the parameter encoded in some evolution, typically a probe system is prepared in a suitable state that then undergoes the evolution. After the parameter is encoded in the probe, a measurement is performed and, based on the outcome statistics, an estimate of the parameter value is assigned.

The frequentist framework is built on the Fisher information (FI), a quantity calculated directly from the likelihood of observing a measurement outcome under the condition that the estimated parameter assumes a given value. Based on the FI, the Cramér-Rao bound (CRB) provides a limit on the precision of an estimation process [6, 7], that is asymptotically saturated by locally unbiased estimators [8]. The maximal attainable FI over all quantum measurements is called the quantum Fisher information (QFI), a quantity that depends only on the estimated evolution and leads to a bound on the highest attainable precision, the quantum CRB [9–12]. Moreover, for unitary evolutions, the QFI and the optimal measurement are independent of the value of the parameter and can be calculated directly from the estimated evolution. This makes the frequentist estimation a powerful framework, that provides a bound on the ultimate precision and at the same time the tools to saturate it asymptotically [13]. The search for the optimal strategy then consists of finding the probe state that maximizes the QFI.

However, there are situations where this framework is too rigid for experimental applications. The optimal measurement provided might not be practical or even experimentally feasible in a given situation. A typical example is the estimation of a rotation in phase space of a continuous-variable system, where the optimal measurement is extremely challenging to implement experimentally. Homodyne detection, on the other hand, is experimentally feasible and strategies using homodyne detection for phase estimation can still saturate the quantum CRB [14–16]. Moreover, it is robust to different forms of noise [17–25].

It is important to note that when the measurement differs from the optimal measurement, the FI with respect to that measurement has to be used and will generally depend on the value of the estimated parameter. The frequentist framework still provides the optimal estimation strategy for the employed measurement, under the assumption that the value of the parameter is sufficiently well known and in the asymptotic limit of many measurements. However, in scenarios where initial information on the parameter is poor or only a limited number of measurement rounds are available, the strategy that maximizes the FI might be far from optimal [26]. In such a case, the Bayesian estimation framework can be employed. In the Bayesian framework, the initial knowledge about the parameter is encoded in a probability distribution

that is updated after a measurement. Here, good probe states lead to a small expected variance of the posterior distribution. Bayesian estimation techniques have been successfully employed for phase estimation [15, 26–30].

In this article, we focus on Gaussian probe states, which have proven useful for phase estimation [14, 15, 31] and generally in quantum metrology [16, 32, 33]. When restricting to single-mode Gaussian probe states and homodyne detection, it is known that in the asymptotic limit of infinitely many measurements, the optimal probe state for a given energy is a squeezed vacuum state [14, 15, 32]. At the same time, a Bayesian analysis of a single measurement round reveals that, for homodyne detection, coherent probe states outperform squeezed states of the same photon number [29]. These two opposite results indicate that the optimal Gaussian probe state for phase estimation with homodyne detection strongly depends on the knowledge of the phase parameter we want to estimate. In this article, we investigate this dependence in more detail, identify the optimal probe states and connect the opposing results of the two extreme cases.

It is structured as follows: In Section 2 we give an overview of the frequentist and Bayesian framework for quantum parameter estimation and in Section 3 we briefly review Gaussian quantum optics. In Section 4 we study the optimal single-mode Gaussian probe states for phase estimation depending on the prior knowledge of the parameter. In Section 5 we look at different strategies for repeated measurements, where the knowledge of the parameter changes between rounds. In Section 6, we compare our findings with those of noisy parameter estimation. Finally, we explicitly calculate the FI for our estimation scenario in Appendix A, include an analysis of the optimal probe states based on the FI including the uncertainty of the estimated parameter in Appendix B and more extensive numerical results of a Bayesian analysis of the estimation problem in Appendix C.

## 2 Quantum parameter estimation

In this section, we present a brief summary of the tools and figures of merit used in the frequentist and Bayesian framework for parameter estimation with quantum systems. For a more detailed treatment of the topic, we refer the reader to Ref. [9–13].

### 2.1 Frequentist estimation

In the frequentist or local framework, it is assumed that the parameter $\theta$ to be estimated has a true value $\theta_0$. Using an estimator $\hat{\theta}$ that is locally unbiased around $\theta_0$, the measurement outcomes $m$ are mapped to an estimate $\hat{\theta}(m)$ of the true value of the parameter $\theta_0$ [34]. The variance $V(\hat{\theta})$ of any locally unbiased estimator of $\theta$ can be lower bounded by the inverse of the Fisher Information (FI), a quantity depending solely on the likelihood $p(m|\theta)$ to observe a given outcome $m$ under the assumption that $\theta$ is the true value of the parameter. The FI reads

$$\mathcal{I}[p(m|\theta)] = \int p(m|\theta) \left[ \frac{\partial}{\partial \theta} p(m|\theta) \right]^2 \mathrm{d}m. \tag{1}$$

The bound $V(\hat{\theta}) \geq 1/\mathcal{I}[p(m|\theta)]$ is referred to as the Cramér-Rao bound (CRB) [6, 7] and can always be saturated in the asymptotic limit by the maximum likelihood estimator [8].

For quantum systems, we assume that the estimated parameter $\theta$ is encoded into a probe state $\rho(\theta)$. Measuring a positive operator-valued measure (POVM) $\{\mathcal{O}_m\}_m$, the likelihood of obtaining the outcome $m$ is given by the Born rule $p(m|\theta) = \mathrm{Tr}(\mathcal{O}_m \rho(\theta))$. Maximizing the FI over all POVMs one obtains the quantum Fisher information (QFI) defined as

$$\max_{\{\mathcal{O}_m\}_m} \mathcal{I}[\mathrm{Tr}(\mathcal{O}_m \rho(\theta))] = \mathcal{I}^Q[\rho(\theta)]. \tag{2}$$

The QFI only depends on the state $\rho(\theta)$ encoding the parameter

$$\mathcal{I}^Q[\rho(\theta)] = \lim_{d\theta \to 0} 8 \frac{1 - \sqrt{\mathcal{F}[\rho(\theta), \rho(\theta + d\theta)]}}{d\theta^2}, \tag{3}$$

where $\mathcal{F}(\rho_1, \rho_2) = \left(\mathrm{Tr}\sqrt{\sqrt{\rho_1}\rho_2\sqrt{\rho_1}}\right)^2$ corresponds to the Uhlmann fidelity between two states. Defining the symmetric logarithmic derivative (SLD) implicitly via the equation

$$S[\rho(\theta)]\rho(\theta) + \rho(\theta)S[\rho(\theta)] = \frac{d}{d\theta}\rho(\theta), \tag{4}$$

one can rewrite the QFI as

$$\mathcal{I}^Q[\rho(\theta)] = 2\,\mathrm{Tr}\left[S[\rho(\theta)]\frac{\partial}{\partial\theta}\rho(\theta)\right]. \tag{5}$$

The variance of the estimator is bounded by $V(\hat{\theta}) \geq 1/\mathcal{I}^Q$. This inequality is called the quantum CRB. The optimal POVM for which the FI equals the QFI is a projective measurement in the eigenbasis of the SLD $S[\rho(\theta)]$ [9, 10].

## 2.2 Bayesian estimation

The Bayesian or global framework does not assume the estimated parameter to have a true value, but treats the value of the parameter as a stochastic variable. The so-called prior distribution $p(\theta)$ captures the initial knowledge of the distribution of this random variable. When new information about the estimated parameter becomes available, the distribution is updated according to Bayes' rule

$$p(\theta|m) = \frac{p(m|\theta)p(\theta)}{p(m)}, \tag{6}$$

where $p(m|\theta)$ is the likelihood to obtain the outcome $m$, assuming that the parameter has value $\theta$, and $p(m)$ is the total probability to obtain the outcome $m$ and is given by

$$p(m) = \int p(m|\theta)p(\theta)\,d\theta. \tag{7}$$

The function $p(\theta|m)$ is the posterior distribution of the estimated parameter and describes the knowledge of the parameter $\theta$ after observing the outcome $m$.

Based on the posterior distribution, the estimator assigns an estimate $\hat{\theta}(m)$ for the parameter $\theta$. In the following, we will use the mean value of the posterior as the estimator, the estimate becomes

$$\hat{\theta}(m) = \int p(\theta|m)\,\theta\,d\theta. \tag{8}$$

The variance of the posterior is defined as

$$V_{\mathrm{post}}(m) = \int p(\theta|m)(\theta - \hat{\theta}(m))^2\,d\theta, \tag{9}$$

and captures the amount of knowledge we have of the parameter $\theta$ and can be used to monitor the estimation progress. To compare different estimation strategies, independently from the actual measurement outcomes, we calculate the average posterior variance (APV)

$$\bar{V}_{\mathrm{post}} = \int p(m)V_{\mathrm{post}}(m)\,dm. \tag{10}$$

So far, all definitions have been introduced for a single measurement round, but the generalization to multiple rounds is straightforward. For repeated measurements, the posterior distribution of one round serves as the prior of the next round.

# 3 Physical setting

In this section, we describe the physical settings of the estimation scenario that we study in this article. We want to estimate the parameter describing a phase rotation acting on a bosonic mode with pure single-mode Gaussian probe states and homodyne detection.

## 3.1 Gaussian probe states

The description of Gaussian states needs some definitions from quantum optics and continuous-variable quantum information, see [35–39] for comprehensive reviews of the topic.

For a bosonic system of a single harmonic oscillator, Gaussian states are states with a Gaussian Wigner function in phase space. They are uniquely determined by the first and second moments of the quadrature operators $\hat{q}$ and $\hat{p}$, that is, the vector of expectation values $\bar{\mathbf{r}} = (\langle\hat{q}\rangle, \langle\hat{p}\rangle)^T$ and the covariance matrix $\sigma$ with entries $\sigma_{ij} = \langle\{\hat{\mathbf{r}}_i - \langle\hat{\mathbf{r}}_i\rangle, \hat{\mathbf{r}}_j - \langle\hat{\mathbf{r}}_j\rangle\}\rangle/2$ [38,39], where $\{A,B\} \equiv AB + BA$ is the anticommutator.

Pure single-mode Gaussian states can be generated by acting consecutively with a squeezing operator $\hat{S}(\xi) = \exp\left(\frac{1}{2}(\xi^* a^2 - \xi a^{\dagger 2})\right)$ and a displacement operator $\hat{D}(\alpha) = \exp\left(\alpha a^\dagger - \alpha^* a\right)$ on the vacuum state $|0\rangle$ and are therefore completely specified by their displacement parameter $\alpha \in \mathbb{C}$, their squeezing strength $r \in \mathbb{R}$, and their squeezing angle $\varphi \in [0, 2\pi)$ [38]. If the squeezing is restricted to a real parameter only, then also a phase rotation

$$\hat{R}(\varphi) = \exp\left(-i\varphi\,\hat{a}^\dagger\hat{a}\right), \tag{11}$$

is needed to describe the most general state $|\alpha, re^{i\varphi}\rangle = \hat{D}(\alpha)\hat{S}(re^{i\varphi})|0\rangle = \hat{D}(\alpha)\hat{R}(\varphi/2)\hat{S}(r)|0\rangle$. Its vector of first moments is given by $\bar{\mathbf{r}} = \sqrt{2}|\alpha|[\cos\tau, \sin\tau]^T$, where $\alpha = |\alpha|e^{i\tau}$, and its covariance matrix reads

$$\sigma = \frac{1}{2}\begin{pmatrix} \cosh 2r - \cos\varphi\,\sinh 2r & \sin\varphi\,\sinh 2r \\ \sin\varphi\,\sinh 2r & \cosh 2r + \cos\varphi\,\sinh 2r \end{pmatrix}. \tag{12}$$

After this, the probe state undergoes the rotation $\hat{R}(\theta)$ we want to estimate, and we end up with the state $\hat{R}(\theta)|\alpha, re^{i\varphi}\rangle = \hat{R}(\theta)\hat{D}(\alpha)\hat{S}(re^{i\varphi})|0\rangle$. This state has the vector of first moments

$$\bar{\mathbf{r}} = \sqrt{2}|\alpha|[\cos(\tau - \theta), \sin(\tau - \theta)]^T, \tag{13}$$

and the covariance matrix

$$\sigma = \frac{1}{2}\begin{pmatrix} \cosh 2r - \cos(2\theta + \varphi)\sinh 2r & \sin(2\theta + \varphi)\sinh 2r \\ \sin(2\theta + \varphi)\sinh 2r & \cosh 2r + \cos(2\theta + \varphi)\sinh 2r \end{pmatrix}. \tag{14}$$

## 3.2 Homodyne detection

Homodyne detection [3, 40] corresponds to performing measurements on a quadrature, for example the position quadrature $\hat{q}$. Thus, the likelihood to obtain outcome $q$ given the phase $\theta$ can be calculated by integrating the Wigner function over the $\hat{p}$-quadrature. The vector of first moments is given by Eq. (13) and the covariance matrix is given by Eq. (14). Accordingly, we find the probability of outcome $q$

$$p(q|\theta) = |\langle q|\hat{R}(\theta)\hat{D}(\alpha)\hat{S}(re^{i\varphi})|0\rangle|^2 = \frac{\exp\left[-\frac{\left(q - \sqrt{2}|\alpha|\cos(\tau - \theta)\right)^2}{\cosh(2r) - \cos(\varphi + 2\theta)\sinh(2r)}\right]}{\sqrt{\pi}\sqrt{\cosh(2r) - \cos(\varphi + 2\theta)\sinh(2r)}}, \tag{15}$$

which is a Gaussian distribution with mean $\mu = \sqrt{2}|\alpha|\cos(\tau - \theta)$ and variance $\sigma^2 = \frac{1}{2}(\cosh(2r) - \cos(\varphi + 2\theta)\sinh(2r))$.

# 4 Optimal probe

In this section, we analyze the estimation process of an unknown rotation in phase space with homodyne detection from a frequentist and a Bayesian perspective and compare the two approaches. In both cases, we look for the optimal pure single-mode Gaussian probe state for a given energy level. In the frequentist framework, the optimal probe state maximizes the Fisher information $\mathcal{I}$, and in the Bayesian framework, the optimal probe state minimizes the APV $\bar{V}_{\text{post}}$.

The likelihood of observing a given measurement outcome $q$ is given by Eq. (15). In preparing the probe state, we have control over the displacement $\alpha = |\alpha|e^{i\tau}$, the strength $r$ and the direction $\varphi$ of the squeezing. The energy $E$ used in the preparation of the probe state depends solely on $|\alpha|$ and $r$ as $E = |\alpha|^2 + \sinh^2 r$. For a fixed energy, the squeezing strength $r$ cannot be chosen independently anymore, but it is determined by the energy $E$ and the amount of displacement $|\alpha|$ by $r = \operatorname{arcsinh} \sqrt{E - |\alpha|^2}$. The parameters $\tau$ and $\varphi$ do not change the energy of the probe state and can always be chosen independently.

## 4.1 Local analysis

The frequentist or local analysis focuses on the optimal strategy for a large number of repeated measurements or, alternatively, on the asymptotic case of having "full" knowledge of the parameter we want to estimate. When estimating a parameter encoded in a unitary evolution, the QFI is independent of the parameter $\theta$ itself, and can be computed using Eq. (3) or (5). In this case it reads

$$\mathcal{I}^Q = 2\sinh^2(2r) + 4e^{2r}(|\alpha|\cos\tau\cos\varphi + |\alpha|\sin\tau\sin\varphi)^2 - 4e^{-2r}(|\alpha|\cos\tau\sin\varphi - |\alpha|\sin\tau\cos\varphi)^2,$$
(16)

and is maximal for $|\alpha| = 0$, in which case $\mathcal{I}^Q = 2\sinh^2(2r)$ [15, 32].

The QFI gives a lower bound to the best achievable precision. We have seen that for a locally unbiased estimator, asymptotically the CRB (and its quantum counterpart) are saturated. However, homodyne detection does not correspond to the optimal measurement given by the SLD in Eq. (4). Therefore, we calculate the FI for pure single-mode Gaussian states with homodyne detection

$$\mathcal{I}(p(q|\theta)) = \frac{2\left(2|\alpha|^2\Sigma\sin^2(\theta - \tau) + \sinh^2(2r)\sin^2(2\theta + \varphi)\right)}{\Sigma^2},$$
(17)

where we have used $p(q|\theta)$ given in Eq. (15) and we have defined $\Sigma \equiv \cosh(2r) - \cos(\varphi + 2\theta)\sinh(2r)$ (see Appendix A for a detailed derivation). The FI is maximized for $|\alpha| = 0$, $r = \operatorname{arcsinh} \sqrt{E}$ and $\varphi = \arccos(\tanh 2r) - 2\theta$ and becomes $\mathcal{I} = 8E(E + 1)$ [15], which is equal to the optimal QFI. This strategy is optimal if the estimate $\hat{\theta}(m)$ is close to the true value of the parameter $\theta$. In Appendix B we analyze bounds on the estimation based on the FI that include the uncertainty of the estimated parameter. Inspired by this asymptotically optimal strategy, we identify a family of strategies that are characterized by allocating all the energy to the squeezing of the probe state, i.e. using squeezed vacuum states. We will refer to this family as the *low uncertainty strategy* (LUS).

## 4.2 Bayesian analysis

Shifting to a Bayesian analysis of the estimation problem, we look for the Gaussian probe state that minimizes the APV. Starting with the likelihood given in Eq. (15), we can calculate the posterior distribution of the phase parameter $\theta$ according to Bayes' law (6) for a given prior distribution. For our analysis, we choose Gaussian distributions as prior. In principle,

any family of distributions can be chosen as priors; Gaussian distributions seem like a natural choice to encode both the estimate and the uncertainty of a variable. Additionally, the Bernstein–von Mises theorem [41, 42] states that in the limit of repeated measurements the posterior distribution converges towards a Gaussian distribution. The choice of prior will influence both the optimal probe state and the estimation. However, the prior should not be regarded as part of the estimation process that can be subjected to optimization, but instead as encoding our knowledge about the parameter prior to the estimation. The mean of the prior distribution can be seen as the estimate of the parameter $\theta$ before the measurement, and therefore we denote it by $\hat{\theta}_0$. The likelihood in Eq. (15) does not depend directly on the mean $\hat{\theta}_0$, but only on the relative angles $\tau - \hat{\theta}_0$ and $\varphi + 2\hat{\theta}_0$. A change in the mean of the prior distribution can be absorbed by a rotation of the probe state. Without loss of generality, we choose $\hat{\theta}_0 = \pi/2$.

Homodyne detection in one quadrature does not provide information on the complementary quadrature. For example, a measurement in $\hat{q}$ cannot distinguish between states that are mirrored by the $q$-axis in phase space. Therefore, we restrict the prior distribution to the interval $[0, \pi)$. By choosing a small enough variance, this can be implemented without artificially restricting the support of the prior. We will limit the values of $\sigma^2$ to the interval $(0, 0.2]$; for $\sigma^2 \leq 0.2$ less than 0.05% of the magnitude of the distribution lies outside of the interval. Finally, averaging over all possible measurement outcomes, we obtain the APV $\bar{V}_{\text{post}}$ according to Eq. (10). Unfortunately, the APV can not be calculated analytically and hence we resort to a numerical analysis of the problem.

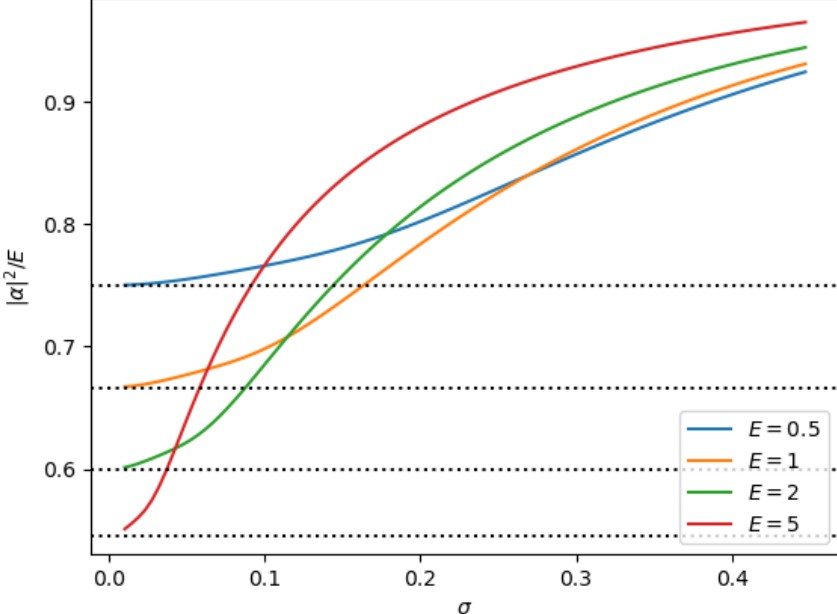

Figure 1: The plot shows the quotient of the energy allocated to the displacement $|\alpha|^2$ and the total energy $E$ of the optimal probe state. This ratio is plotted against the square root of the variance $\sigma$ of the prior distribution of the phase for different energies: $E = 0.5$ in blue, $E = 1$ in orange, $E = 2$ in green, and $E = 5$ in red. The black dotted lines show the optimal ratio $(E + 1)/(2E + 1)$ for a frequentist analysis of the HUS.

If the variance of the prior distribution is sufficiently large, our numerical results consistently indicate a one-parameter family of states as optimal, see Appendix C for the numerical results. These states are characterized by $\tau = \hat{\theta}_0 \pm \pi/2$ and $\varphi = -2\hat{\theta}_0$, where the estimate $\hat{\theta}_0$ is the mean of the prior distribution. The last parameter $|\alpha|$ depends on the energy of the probe state and the prior distribution. Fig. 1 shows the optimal allocation of energy to the displacement $|\alpha|^2/E$ of the probe, depending on the variance of the prior $\sigma^2$ for different values of the total energy $E$ of the probe.

There is a plausible explanation for why this family of states is optimal. The sensitivity of homodyne detection in the $\hat{q}$-quadrature with respect to the parameter $\theta$ is maximal at $q = 0$. This can be seen directly from Eq. (15), as $\cos(\tau - \theta)$ changes most rapidly at $\tau = \theta \pm \pi/2$. Choosing $\tau = \hat{\theta}_0 \pm \pi/2$, we expect this to be the case according to the prior distribution and the probe state to be rotated onto the $p$-axis, i.e., to have the first moment vector $\bar{\mathbf{r}} = \sqrt{2}|\alpha|(0, \pm 1)^T$. With the second condition $\varphi = -2\hat{\theta}_0$, we expect the probe state to be squeezed along the $\hat{q}$-quadrature after it has undergone the unknown phase rotation. With this choice, the Wigner function of the state $\hat{R}(\theta)\hat{D}(\alpha)\hat{R}(\varphi/2)\hat{S}(r)|0\rangle$ is elongated along the $\hat{p}$-quadrature, which minimizes the uncertainty in the $\hat{q}$-quadrature. Since a homodyne measurement informs us of the value of the quadrature $\hat{q}$, this seems a reasonable choice. We will refer to the strategy using only states for which $\tau = \hat{\theta}_0 \pm \pi/2$ and $\varphi = -2\hat{\theta}_0$ as *high uncertainty strategy* (HUS). Fig. 2 shows the ratio of the APV and the prior variance against the prior variance for different probe states of the HUS, including the optimal probe depending on the prior variance.

## 4.3 Comparison between Bayesian and local approach

We have seen that the optimal pure single-mode Gaussian probe state for the estimation of an unknown phase $\theta$ depends on the knowledge or uncertainty of the estimated parameter. A local analysis of the problem suggests squeezed vacuum states that allocate all the available energy into squeezing as optimal probe states. For squeezed vacuum states the uncertainty ellipse in phase space is elongated in one quadrature and squeezed in the other. Among states of a given energy, squeezed vacuum states have the smallest uncertainty in the appropriate quadrature. This makes them universally powerful probe states for local estimation [16]. Consequently, the outcome distribution is very sensitive to the parameter $\theta$ and locally saturates the Quantum Cramér-Rao bound [15]. However, the high uncertainty in the complementary quadrature makes squeezed states quickly become worse probe states if the estimated parameter $\theta$ is only known up to some limited precision. This fact is also reflected in the peaked shape of the FI in Fig. 5 in Appendix B. It is therefore not surprising that a Bayesian investigation of the problem finds that for a high uncertainty of the parameter $\theta$, more energy should be used to displace the probe state in phase space.

Before comparing the two strategies, we first analyze the HUS in the local framework and the LUS in the Bayesian framework. Setting $\varphi = -2\theta$ and $\tau = \theta - \pi/2$,[1] we can calculate the Fisher information for these probe states and obtain

$$\mathcal{I} = 4|\alpha|^2 e^{2r}. \tag{18}$$

For a fixed total energy $E = |\alpha|^2 + \sinh^2 r$, this expression is maximized for $|\alpha|^2 = E(E+1)/(2E+1)$ and $r = \log(2E + 1)/2$, resulting in $\mathcal{I} = 4E(E + 1)$. In Fig. 1 we see how the optimal probe states calculated with a Bayesian analysis approach the optimal probe states calculated with the local analysis for decreasing values of the variance of the prior distribution, both within the HUS. Numerically calculating the APV for the states of the LUS, we find that for a small enough variance of the prior also a Bayesian analysis identifies squeezed vacuum states as the

---

[1] We are not writing $\hat{\theta}(m)$ because in the asymptotic limit the estimate is equal to the real value of $\theta$.

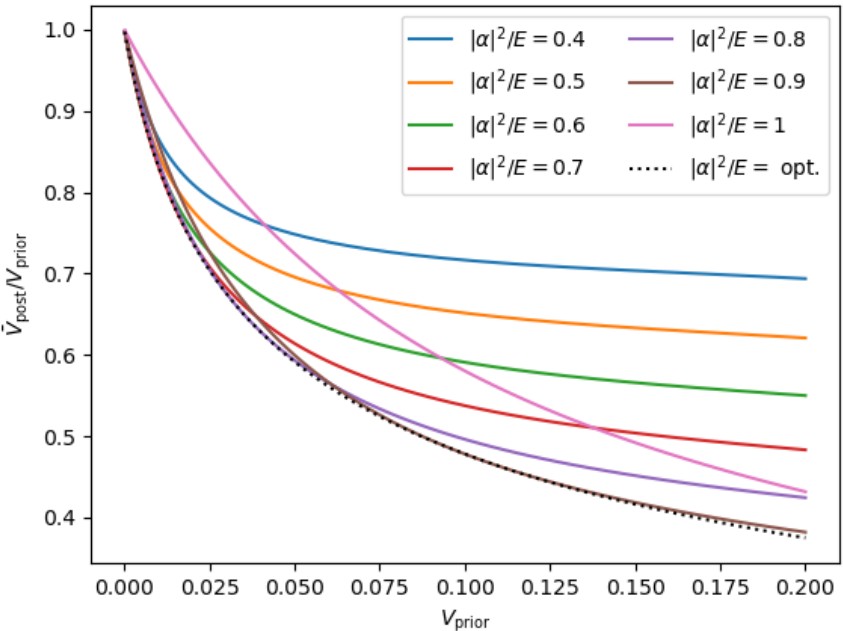

Figure 2: The plot shows the ratio of the average posterior variance to the prior variance plotted against the variance of the prior distribution for the HUS with $\varphi = -\pi$, $\tau = 0$, and $E = 2$. In blue, yellow, green, red, violet, brown, and pink we see strategies with a fixed allocation of energy to displacement, going from $|\alpha|^2/E = 0.4$ to $|\alpha|^2/E = 1$ in steps of 0.1. The green line coincides with the optimal frequentist probe state $|\alpha|^2/E = (E+1)/(2E+1) = 0.6$ within the HUS. The black dotted curve corresponds to the optimized probe of the HUS, where the optimal allocation of energy to displacement and squeezing of the probe state depends on the prior variance $\sigma^2$.

optimal probe states, see Fig. 3. Nonetheless, the optimal angle $\varphi$ of the probe states still differs between the two approaches. Fig. 6 in Appendix C shows the optimal angle $\varphi + 2\hat{\theta}_0$ depending on the prior variance $\sigma^2$ for different values of the total energy $E$ of the probe state (continuous lines). The optimal angle $\varphi$ is quite sensitive in a change of the variance and only converges toward the optimal angle $\varphi = \arccos(\tanh 2r) - 2\hat{\theta}_0$ obtained in the local analysis in section 4.1 (dotted lines) when the variance approaches zero. This behavior likely stems from an unequal trade-off between over- and underestimating the phase, also reflected by the asymmetric shape of the peak in Fig. 5.

Performing a detailed analysis of the estimation problem, we can identify the optimal probe state in the 3-dimensional parameter space $(|\alpha|, \tau, \varphi)$ of pure single-mode Gaussian states with a fixed level of energy $E$. Interestingly, we find two distinct optimal strategies of families of states. The first strategy is based on a local analysis and consists of employing squeezed vacuum states with $\alpha = 0$. The only free parameter $\varphi$ depends on the energy $E$ of the probe state and the prior distribution of $\theta$ (see Appendix C and Fig. 6 for the dependence). The second strategy is obtained through a Bayesian analysis of the problem and results optimal when one has little knowledge of the value of $\theta$. It is characterized by $\tau = \hat{\theta}_0 - \pi/2$ and $\varphi = -2\hat{\theta}_0$. The free parameter $|\alpha|$ depends on the energy $E$ of the probe state and the prior distribution of $\theta$ as seen in Fig. 1.

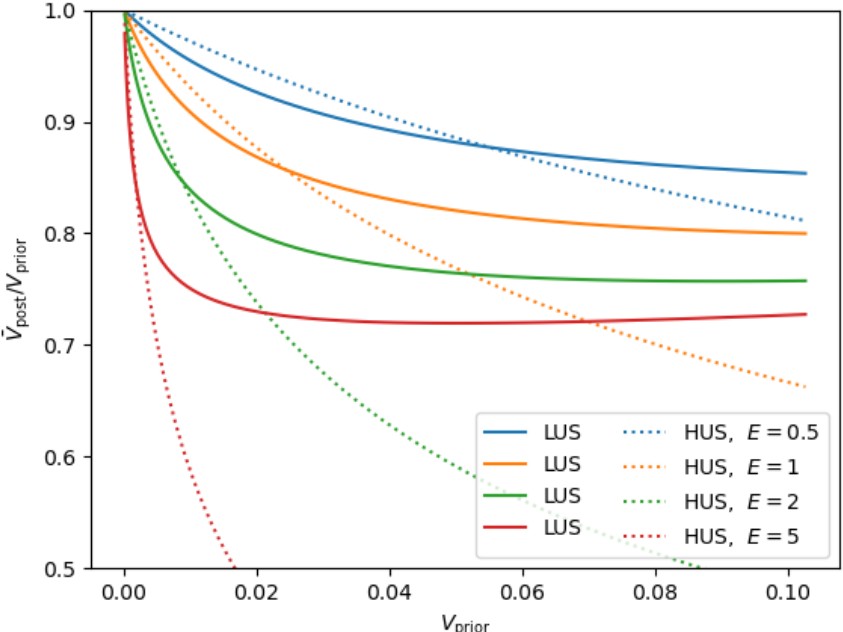

Figure 3: The plot shows the ratio of the average posterior variance to the prior variance plotted against the variance of the prior distribution. Dotted lines correspond to the HUS with $\varphi = -\pi$, $\tau = 0$, and optimal allocation of energy to displacement $|\alpha|^2$ and squeezing strength $r$ of the probe. Continuous lines correspond to the LUS with $|\alpha|^2 = 0$ and optimal angle $\varphi$. The different energies of the probe states are shown in blue for $E = 0.5$, in orange for $E = 1$, in green for $E = 2$, and in red for $E = 5$. We see that for larger variances the HUS has the best performances, whereas, for small prior variances, the situation is reversed. The point where the optimal strategy changes depends on the energy of the probe. For a smaller energy, this point is reached for larger variances than for probes with higher energy.

The FI of the LUS with $\mathcal{I} = 8E(E + 1)$ is twice as large as the largest FI for the HUS. This strategy is optimal when the variance of the prior distribution is small. Fig. 3 compares the two strategies. While for larger variances of the prior distribution the HUS clearly outperforms the LUS, for small variances the situation is reversed. The value of $\sigma^2$ where this change occurs depends on the energy of the probe states. Remarkably, there does not exist a continuous path in parameter space describing the optimal strategy, but there exists a clear cut between the two presented strategies. Both strategies are local minima in the parameter space $(|\alpha|, \tau, \varphi)$. A numerical minimization of the APV consistently results in one of the two strategies, depending on the starting point of the minimization in the parameter space. An example is given in Fig. 7 in Appendix C, the different lines stem solely from different starting points of the numerical optimization. Which of the two strategies also corresponds to the global minima depends on the energy of the probe and the prior distribution of $\theta$.

# 5 Repeated measurements

So far, we have only considered single measurement rounds. When extending our analysis to repeated measurements, it is important to bear in mind that the optimal probe state depends on the knowledge of the estimated parameter, which in turn changes after each measurement. A rigid strategy that uses the same fixed probe state (and measurement) for multiple measurement rounds cannot capitalize on the knowledge gained. In the following, we will look at increasingly sophisticated, but in turn impractical or costly, strategies, benefiting from the increasingly more precise estimate of the phase.

We have seen that the optimal probe state strongly depends on the estimate $\hat{\theta}(m)$ at a given point in the estimation. Using only this information, the measurement procedure can be substantially improved [4,5,28,43]. Adapting for the estimate can be implemented efficiently by a global phase on the probe state or equivalently by a rotation of the homodyning. Although this strategy adapts the estimate, the optimal probe state minimizing the APV depends on the prior distribution of the estimated parameter. A strategy that calculates the optimal probe state in each round will on average outperform any other strategy. However, it might be challenging to implement such a strategy, since the optimal probe state has to be numerically determined on the fly based on the current prior distribution.

In the scenario of estimating an unknown rotation in phase space with pure single-mode Gaussian states and homodyne detection, we have already determined the optimal probe states for Gaussian prior distributions, depending on the mean and variance. The posterior distribution after one measurement will not be a Gaussian distribution itself, as the parameter $\theta$ is encoded both in the mean and in the variance of the likelihood of Eq. (15). However, the Gaussian prior distribution leads to a distribution that is close to a Gaussian distribution, such that the optimal probe state is nearly identical. Additionally, by the Bernstein-von Mises theorem, we know that the posterior distributions tend toward a Gaussian distribution for a large number of rounds. Assuming a Gaussian distribution with the same mean and variance encoding our knowledge about the parameter at a given point of the estimation simplifies the protocol considerably. In such a case, the optimal probe state does not need to be calculated in each round, but the optimization can be run "a priori" of the estimation process, e.g. Fig. 1 and 6 (Appendix C). During the estimation process, the mean and variance of the posterior distribution are calculated after each measurement round. These are then used to determine the optimal probe state without the need to run an optimization over all probe states.

We can even make a further simplification: Instead of using the variance of the posterior distribution at a given point in the estimation process, we can use the expected variance of the posterior distribution at this point in the estimation. The optimal probe state is then chosen according to the estimate and the expected variance of the prior distribution at that point. Depending on the measurement setup, this strategy might be more feasible to implement experimentally. The probe state still varies between different measurement rounds, but does so according to a predetermined rule. Only the phase needs to be adapted to the measurement outcomes, which can be implemented with a global rotation of the probe state or, alternatively, with a rotation of the quadrature of the measurement.

We compare the different strategies described above for probe states chosen according to the HUS. The optimal probe state of the HUS depends on the prior distribution. Assuming a Gaussian distribution, the mean $\hat{\theta}_0$ determines the angles $\tau = \hat{\theta}_0 - \pi/2$ and $\varphi = -2\hat{\theta}_0$ of the probe state (or equivalently the measurement direction [5]). The variance determines the optimal allocation of energy to the displacement $|\alpha|$ and the squeezing strength $r$ of the probe according to Fig. 1.

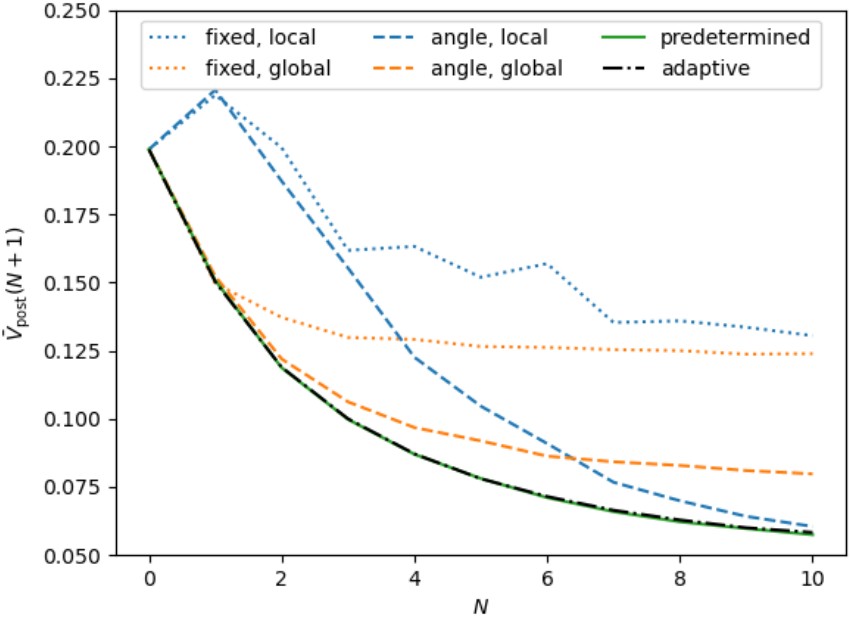

Figure 4: The plot shows the average posterior variance $\bar{V}_{\text{post}}$ obtained for 1000 trajectories vs the number of rounds $N$ for the HUS. To better see the differences in the variances for larger rounds, we multiply the variance with $(N+1)$. We have chosen the energy $E = 2$ and the prior variance $V_{\text{prior}} = 0.2$. Dotted lines correspond to the strategies in which the state and the measurement are fixed and dashed lines correspond to strategies in which the distribution of energy is fixed but the measurement direction changes at each round according to the estimate. In blue, the displacement is chosen according to a local analysis $\alpha^2/E = (E+1)/(2E+1)$ for small variances, in orange, according to a Bayesian analysis based on the initial distribution. The solid green line represents the predetermined strategy in which the distribution of energy is optimized for the expected average posterior variance of the prior and the black dash-dotted line represents the fully adaptive strategy.

In Fig. 4, we see the performance of fixed probe states as dotted lines. Probe states where only the angle is adapted to the changing estimate, but the allocation of energy does not change, are shown as dashed lines. For the blue lines, the optimal probe state within the HUS is chosen for the asymptotic limit of repeated measurements, whereas the orange lines show the optimal probe state according to the initial variance. The continuous green line corresponds to the predetermined strategy, where the angle of the probe is updated according to the estimate, and the allocation of energy in the probe state changes, but follows a predetermined rule. Finally, the black dash-dotted line shows the adaptive strategy, where in each step the probe state corresponds to the optimal probe state for a Gaussian distribution with the same mean and variance. For computational reasons, we cannot display the optimal strategy, as calculating the optimal probe state in each step is very inefficient. We see that fixed strategies (dotted lines) perform poorly with respect to other strategies. Choosing the optimal probe state according to the initial distribution without updating it (orange lines) performs well initially, but is soon outperformed by the optimal probe state in the asymptotic limit. As expected, changing the probe state in each round gives the best performance. Surprisingly, adapting based on the variance does not perform better than just changing the probe state based on the expected variance at that point.

## 6 Connection to noisy estimation

In this section, we compare the results presented in this paper with those obtained in noisy parameter estimation [17, 18, 20–22, 44]. In a noisy estimation scenario, in addition to the estimated evolution, the system is exposed to a noise channel, such as phase diffusion, thermal photon noise, and photon loss. Noise is detrimental to parameter estimation and generally will change the optimal probe state for the estimation. If the noise commutes with the estimated evolution, the effect of the noise on the estimation can be compared to an increase in the uncertainty of the parameter.

A typical example of noisy phase estimation is analyzed in [17], where the noise is described by a phase diffusion with noise amplitude $\Gamma$, and the equation of motion reads

$$\frac{\partial \rho}{\partial t} = \Gamma \mathcal{L}[a^\dagger a]\rho, \tag{19}$$

where $\mathcal{L}$ represents the dissipator $\mathcal{L}[\mathcal{O}]\rho = 2\mathcal{O}\rho\mathcal{O}^\dagger - \mathcal{O}^\dagger\mathcal{O}\rho - \rho\mathcal{O}^\dagger\mathcal{O}$. When estimating the phase rotation $\theta$ with a pure Gaussian probe state $|\psi\rangle$, the state of the system after time $t$ reads

$$\rho_\theta(t) = \mathcal{N}_\Delta\left(\hat{R}(\theta)|\psi\rangle\langle\psi|\hat{R}^\dagger(\theta)\right) = \hat{R}(\theta)(\mathcal{N}_\Delta |\psi\rangle\langle\psi|)\hat{R}^\dagger(\theta), \tag{20}$$

where $\hat{R}(\theta) = \exp(-i\theta a^\dagger a)$, and $\mathcal{N}_\Delta = e^{\Delta\mathcal{L}}$ represents the noise channel with $\Delta \equiv \Gamma t$. The last equality is obtained because $\hat{R}(\theta)$ and the channel commute [17]. Phase-diffusive noise destroys the off-diagonal elements of the density operator in the Fock basis and with this the phase information carried by the state. This makes phase-diffusive noise the most destructive noise for phase estimation [17]. The problem can then be rephrased as estimating the phase $\theta$ for a family of input states of the form $\mathcal{N}_\Delta |\psi\rangle\langle\psi|$. Ref. [17, 18] investigated the optimal Gaussian probe states for noisy phase estimation within a frequentist framework, that is, they looked for the state that maximizes the QFI. They found that the noisier the dynamics, the more beneficial it is to increase the proportion of energy in the displacement of the probe.

Although in a different framework, those results are in line with our results. In Fig. 3, we can see how the LUS, where all the energy is put into the squeezing of the probe state, quickly starts to become suboptimal as soon as the prior variance increases. In Fig. 1, we see that for a large prior variance, the optimal probe state has more energy in the displacement until the optimal probe state is almost coherent.

These qualitatively similar results are not surprising. We have seen that the performance of different probe states depends on our ability to predict the state of the system after it undergoes the transformation that we want to estimate. Introducing some noise of the form $\mathcal{N}_\Delta |\psi\rangle\langle\psi|$ leads to an increase in the uncertainty of $\theta$. Hence, for the preparation of the optimal probe state, noise can be seen as a lack of knowledge of the parameter $\theta$ [20]. In summary, the strategies presented here may also be useful for noisy estimation scenarios where the noisy channel commutes with the dynamics.

## 7 Conclusions

This work investigates the role of knowledge-dependent probe design in quantum metrology for practical and noise-resistant phase estimation techniques. We identify the optimal pure single-mode Gaussian probe state for the estimation of an unknown phase rotation with homodyne detection. In a Bayesian framework, our findings indicate that the probe state that minimizes the average posterior variance of the estimated parameter significantly changes with the initial knowledge we have of the parameter itself. We identify two different strategies that

are optimal, depending on the variance of the prior distribution encoding our knowledge of the parameter. If only a fixed amount of energy is available for the preparation of the probe state, our analysis suggests that for high uncertainty, it is better to prioritize displacing the probe state in phase space. In the limit of having little initial information on the parameter, we recover the results presented in Ref. [29] for uniform priors. In contrast, for increasingly more precise estimates of the parameter, it becomes beneficial to prioritize squeezing the probe state until we approach the asymptotically optimal strategy introduced in Ref. [15]. Surprisingly, this is not a smooth transition, but there is a sudden jump from squeezed displaced states to squeezed vacuum states.

We then look at repeated measurements. Since after each measurement our knowledge of the parameter changes, so does the optimal probe state. We emphasize that the optimal probe state does not only depend on the estimate of the parameter, but on the distribution encoding our knowledge of the parameter itself. We investigate different strategies that adapt to the changing amount of information available to various degrees and compare those strategies. Strategies that use the same probe state independent of the previous measurement outcomes are generally outperformed by those that adapt to the estimate. The optimal strategy calculates the optimal probe state after each measurement round, but since this procedure becomes impractical, we propose two simpler alternatives: adapting the probe state according to the variance of the distribution and to the expected variance of the distribution for a given measurement round.

We compare our results to those of noisy phase estimation and find qualitatively similar results: the less we are able to predict the state of the probe after the transformation, either due to a lack of knowledge of the parameter or due to a more noisy estimation, the better it is to move away from the asymptotically optimal, but unstable, squeezed vacuum probe states and towards more stable coherent probe states.

In our work, we have connected the findings from a single-shot Bayesian analysis of phase estimation with the asymptotic limit of a frequentist analysis. We are aware that in many realistic estimation scenarios the number of probes will be high enough to approximate the asymptotic limit. However, estimation problems with a limited number of probe states available are becoming more relevant for quantum error correction in quantum computing and quantum networks [45, 46]. Additionally, when the parameter to be estimated is itself a random variable sampled from some distribution or changes over time, the knowledge about the parameter may always be limited.

## Acknowledgments

The authors are thankful to Mohammad Mehboudi, Michalis Skotiniotis, Nicolai Friis and Géza Tóth for helpful discussions.

**Funding Information**    R.R.R. is financially supported by the Ministry for Digital Transformation and of Civil Service of the Spanish Government through the QUANTUM ENIA project call - Quantum Spain project, and by the EU through the RTRP - NextGenerationEU within the framework of the Digital Spain 2026 Agenda. S.M. acknowledges financial support from the Austrian Federal Ministry of Education, Science and Research via the Austrian Research Promotion Agency (FFG) through the project FO999921407 (HDcode) funded by the European Union—NextGenerationEU.

# A  Fisher information for homodyne measurements on Gaussian states

When measuring in a continuous POVM with outcomes $m$, the Fisher information is defined as

$$\mathcal{I}(p(m|\theta)) = \int_{-\infty}^{\infty} \frac{1}{p(m|\theta)}\left(\frac{\partial p(m|\theta)}{\partial \theta}\right)^2 \mathrm{d}m. \tag{A.1}$$

In the case of homodyne detection in the position quadrature of Gaussian states, the likelihood of observing a given outcome $q$ is given by

$$p(q|\theta) = |\langle q|\hat{R}(\theta)\hat{D}(\alpha)\hat{S}(re^{i\varphi})|0\rangle|^2 = \frac{\exp\left[-\frac{(q-\sqrt{2}|\alpha|\cos(\tau-\theta))^2}{\cosh(2r)-\cos(\varphi+2\theta)\sinh(2r)}\right]}{\sqrt{\pi}\sqrt{\cosh(2r)-\cos(\varphi+2\theta)\sinh(2r)}}. \tag{A.2}$$

When estimating the phase $\theta$, we have

$$\frac{\partial p(q|\theta)}{\partial \theta} = \frac{\left(2\sqrt{2}|\alpha|\Sigma(\mu-q)\sin(\theta-\tau)+\sinh(2r)\sin(2\theta+\phi)\left(2(q-\mu)^2-\Sigma\right)\right)}{\sqrt{\pi}\Sigma^{5/2}}e^{-\frac{(q-\mu)^2}{\Sigma}}, \tag{A.3}$$

where we define $\mu \equiv \sqrt{2}|\alpha|\cos(\tau-\theta)$ and $\Sigma \equiv \cosh(2r)-\cos(\varphi+2\theta)\sinh(2r)$. Finally, we calculate the FI

$$
\begin{aligned}
&\mathcal{I}(p(q|\theta))\\
&= \int_{-\infty}^{\infty} \frac{\left(2\sqrt{2}|\alpha|\Sigma(q-\mu)\sin(\theta-\tau)+\sinh(2r)\sin(2\theta+\varphi)\left(\Sigma-2(q-\mu)^2\right)\right)^2}{\sqrt{\pi}\Sigma^{9/2}}e^{-\frac{(q-\mu)^2}{\Sigma}}\mathrm{d}q\\
&= \frac{2\left(2|\alpha|^2\Sigma\sin^2(\theta-\tau)+\sinh^2(2r)\sin^2(2\theta+\varphi)\right)}{\Sigma^2},
\end{aligned} \tag{A.4}
$$

obtaining the expression of Eq. (17).

# B  Optimal probe states – FI analysis

Previously, we calculated the optimal probe state in the asymptotic limit of repeated measurements. However, the FI can also be used to analyze the suitability of probe states including the uncertainty of the estimated parameter. The Van Trees bound [47] is often seen as the Bayesian analog to the CRB and lower bounds the APV as

$$\bar{V}_{\text{post}} \geq \frac{1}{\mathcal{I}[p(\theta)]+\bar{\mathcal{I}}[p(m|\theta)]}, \tag{B.1}$$

where $\bar{\mathcal{I}}[p(m|\theta)] = \int \mathcal{I}[p(m|\theta)]p(\theta)\mathrm{d}\theta$ is the average FI of the likelihood $p(m|\theta)$, and $\mathcal{I}[p(\theta)]$ is the FI of the prior and is computed according to Eq. (1). Similarly to the CRB, also the Van Trees bound has a quantum version [13] that reads

$$\bar{V}_{\text{post}} \geq \frac{1}{\mathcal{I}[p(\theta)]+\mathcal{I}^Q[\rho(\theta)]}, \tag{B.2}$$

where the first summand in the denominator is the same as before but $\mathcal{I}^Q[\rho(\theta)]$ is now the QFI of the quantum state $\rho(\theta)$. The bound can be obtained using the fact that $\mathcal{I}^Q[\rho(\theta)] \geq \mathcal{I}[p(m|\theta)]$. Moreover, if we deal with estimation protocols in which the QFI does

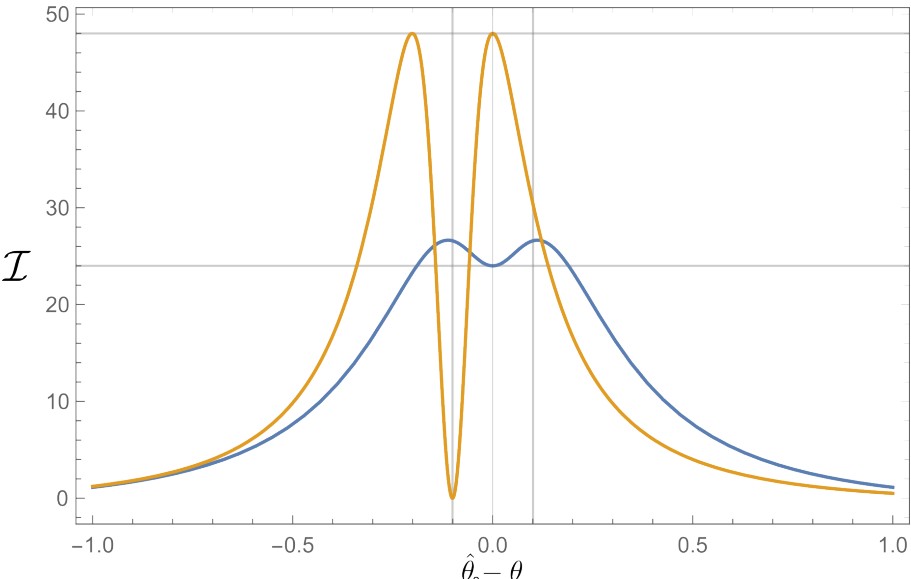

Figure 5: The plot shows the Fisher information depending on the difference $\hat{\theta}_0 - \theta$ for $E = 2$. In blue we see the state of the HUS maximizing the FI and in orange the state of the LUS with the largest FI. Horizontal gray lines show the FI for the two strategies at the point $\hat{\theta}_0 = \theta$. Vertical lines show the angle $\pm\arccos\bigl(\tanh(2\arcsin\sqrt{E})\bigr)/2$, corresponding to the optimal tilting of the squeezed vacuum probe state after the rotation. The case $\hat{\theta}_0 - \theta = -\arccos\bigl(\tanh(2\arcsin\sqrt{E})\bigr)/2$ results in a horizontally squeezed state after the phase rotation, where $\mathcal{I}[p(m|\theta)] = 0$. For $\hat{\theta}_0 - \theta < -\arccos\bigl(\tanh(2\arcsin\sqrt{E})\bigr)/2$ the FI rises again, but leads to a wrongful estimation.

not depend on the parameter $\theta$ (e.g. when the parameter is encoded in a unitary evolution), we have $\bar{\mathcal{I}}[p(m|\theta)] \leq \int \mathcal{I}^Q[\rho(\theta)]p(\theta)\mathrm{d}\theta = \mathcal{I}^Q[\rho(\theta)]$ and we get Eq. (B.2) as a stronger bound than Eq. (B.1).

We have seen that the quantum FI for phase estimation with Gaussian states is optimized by squeezed vacuum states and scales with the energy as $\mathcal{I}^Q[\rho(\theta)] = 8E(E+1)$. Using this and the quantum Van Trees bound we immediately obtain a lower bound on the APV

$$\bar{V}_{\text{post}} \geq \frac{1}{\frac{1}{\sigma^2} + 8E(E+1)}, \tag{B.3}$$

where $\sigma^2$ corresponds to the variance of the prior distribution. This is just a lower bound and is optimized over all possible POVMs. To calculate the Van Trees bound we need the average FI. We can bound the average FI by the maximal FI for our problem. But, as we have seen, the FI for squeezed vacuum is also $8E(E+1)$. So to actually improve the lower bound we have to calculate the average FI for a given state and optimize over all possible states of a given energy. The optimal average FI for a given variance of the prior can not be calculated analytically and would require a numerical exploration of the problem. We refrain from this exercise here, as calculating a lower bound on the APV when we have already calculated the actual APV is not very meaningful.

Nonetheless, we can calculate the FI of some probe states to better understand the problem. Fig. 5 shows the FI of the two states of each strategy that maximize the FI. Recall that the state of the HUS that maximizes the FI has $|\alpha|^2 = E(E+1)/(2E+1)$, $\tau = \hat{\theta}_0 - \pi/2$, $r = \log(2E+1)/2$, and $\varphi = -2\hat{\theta}_0$ and the state of the LUS has $\alpha = 0$, $r = \operatorname{arcsinh}\sqrt{E}$, and

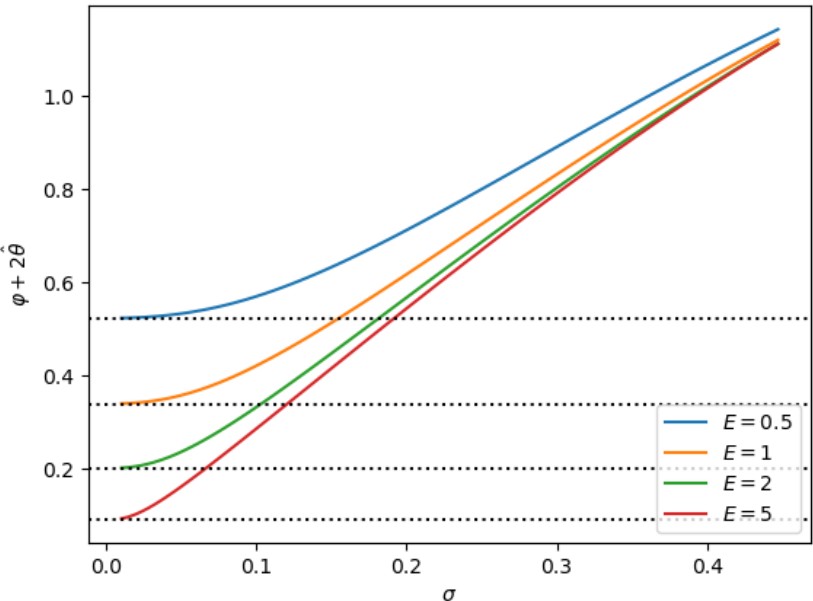

Figure 6: The plot shows the optimal angle $\varphi$ for the LUS plotted against the square root of the variance $\sigma$ of the prior distribution of the phase for different energies: $E = 0.5$ in blue, $E = 1$ in orange, $E = 2$ in green, and $E = 5$ in red. The black dotted lines show the optimal angle $\arccos\left(\tanh\left(2\operatorname{arcsinh}\sqrt{E}\right)\right)$ for a frequentist analysis of this strategy.

$\varphi = \arccos(\tanh 2r) - 2\hat{\theta}_0$. The FI is plotted against the difference between the estimate and the actual angle $\hat{\theta}_0 - \theta$. We see that while the FI is large for the LUS-state at $\hat{\theta}_0 = \theta$, it is quite peaked and falls off quickly. In contrast, the FI of the optimal HUS state is only half as large, i.e. $4E(E+1)$, for $\hat{\theta}_0 = \theta$, and even increases when $\theta$ moves away from the estimate. There is also another important fact to note. While it is true that the FI has a second peak for the LUS-state, homodyne detection cannot distinguish squeezed vacuum states from the same state mirrored on the $\hat{p}$ axis in phase space. Therefore, the second peak is misleading, the FI might be high in this region of $\theta$, but the estimation actually identifies a wrong angle $\theta$. This reminds us of the fact that an analysis based purely on the FI can lead to wrong results about optimality when the uncertainty of the parameter is not sufficiently small.

## C   Optimal probe states – Bayesian analysis

When restricting to probe states with $\alpha = 0$, we can optimize the angle $\varphi$ so that the APV is minimized. In Fig. 6, the optimal angle is shown depending on the variance of the prior Gaussian distribution. In the limit of small variances, this recovers the optimal angle $\varphi = \arccos\left(\tanh\left(2\operatorname{arcsinh}\sqrt{E}\right)\right)$ obtained in the local analysis.

We extensively searched the parameter space $(|\alpha|, \tau, \varphi)$. Fig. 7 shows the optimal probe states depending on the variance of the prior distribution for the energy of the probe state $E = 0.5$ and how well they perform. Interestingly, two local minima are found, depending on the starting point of the optimization in parameter space. These coincide with the two strategies introduced before, except for small variances where numerical imprecisions affect the optimization.

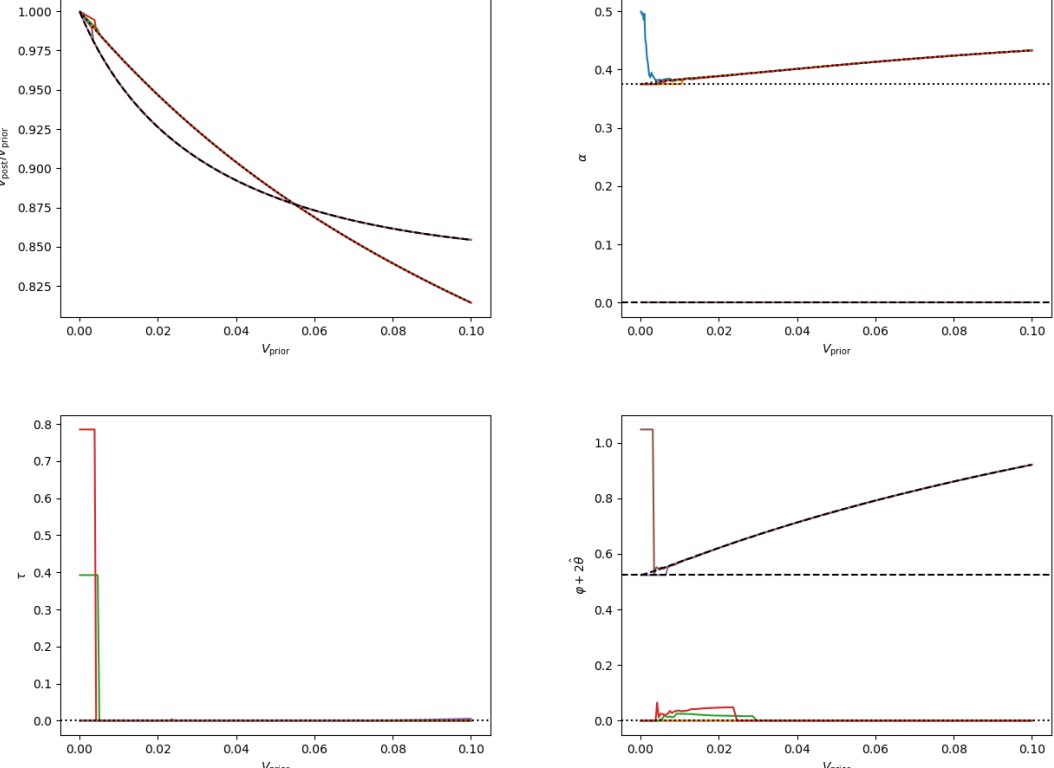

Figure 7: The plots show the ratio between the average posterior variance and prior variance and the parameter $|\alpha|^2$, $\tau$ and $\varphi$ for the probe states optimized over the full parameter space and energy $E = 0.5$. Different lines show different starting points of the optimization. We see two distinct local minima, where the optimization ends up depending on the starting point. We clearly recognize the two strategies discussed earlier. As a reference, the HUS is shown as black dotted line and the LUS as black dashed line.

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
