# Peer review of "Knowledge-dependent optimal Gaussian strategies for phase estimation"

_SciPost Physics, doi:SciPost Phys. 18, 167 (2025)_

## Round 2 · Referee Report · Robert Trenyi (Referee 1) · 2025-3-27

Report
The authors also investigate the case of repeated measurements, where the best strategy is to update probe states after each measurement round. For practicality, they also consider simplified versions of the fully adaptive case, which can be almost as good as the fully adaptive strategy.
A noisy scheme with phase diffusive noise as an example is also considered. Their method can be utilized as this type of noise can be thought of as having a higher prior uncertainty in the parameter, therefore it is better to put more energy in displacing the probe state.
As far as I know these results are new and can be of interest to the quantum information community, especially in designing metrological experiments with continuous variable systems.
The manuscript is sufficiently self-contained, whenever it cannot be, then the proper references are provided. The manuscript is well written and logically structured, so it can be followed easily. The figures are easy to grasp and informative. Moreover, the manuscript has a clear message, calculations that seem correct, even though I did not check all of them. I only checked till Eq. (14) and the derivation of (A3) in Appendix A. All of the analytical calculations seem reproducible.
Based on the above, I strongly recommend the publication of the manuscript in SciPost Physics.
I have the following suggestions/questions, and also found some typos:
• At the end of the introduction there should be a brief sentence about the content of the appendices.
• Regarding the previous point I did not find direct references to Appendix B, C in the main text (apart from referring to Fig. 6 that is in Appendix C), even though it contains useful information.
• 6th line of the 2nd paragraph of III.A.: The vector r should contain (<q>,) instead of (<q>,<q>)?
• It would be useful to define the anticommutator {,} when defining the covariance matrix in the second paragraph of III.A.
• „In this section, we describe the estimation scenario that we are looking at.” sounds too informal to me (Section III. beginning)
• Can the authors give a simple intuitive argument already at the beginning of IV.C. why is it better to allocate all the energy in the squeezing of the state for the frequentist approach?
• III.B: typo in title homodye -> homodyne
• In Eqs. (14) and (16) the entries of the matrices should be better spaced for readability, if possible, in this two-column format
• What is the intuitive reason that the parameter \varphi changes significantly between the two approaches (observation of IV.C)?
• Is there a particular reason why \varphi=2\pi-2\theta is used instead of the previously used \varphi=-2\theta in the penultimate paragraph of Section IV.C?
• For Eq. (16), it should be stated which equation is evaluated from II.A for clarity.
• Do I understand correctly that the authors claim that there cannot be other optimal strategies apart from HUS and LUS based on numerical evidence?
• What is exactly the difference between the “simplified” and the “pushed further” strategies in Section V? The authors should clarify this a bit more in my opinion.
• Would it be possible to obtain meaningful results for the case when the noise does not commute with the dynamics? Maybe the authors can comment on that.
• 1st reference: Americal Journal ->American Journal
• Just out of curiosity: Are there results for general non-Gaussian states? What can they achieve in each metrological setting?
• Also out of curiosity: Do similar results also hold for other types of measurements (like heterodyne)?
Recommendation
Publish (meets expectations and criteria for this Journal)

---

## Round 2 · Referee Report · Anonymous (Referee 2) · 2025-4-4

Strengths
- Very well written paper, with a clear logical structure and well explained theoretical background.
- Clear purpose and results, which are presented in a structured and concise manner.
- Addresses relevant topics.
Weaknesses
Report
The results presented are intuitive, especially following the motivation provided by the authors and as such the results of the paper make sense to me. The detailed analysis provided gives an interesting insight into the choices necessary for performing parameter inference, depending on the type of estimation we occur. Moreover, it is easy to see how the theory, albeit for a relatively simple system, would be useful for more complex tasks also.
Overall, the paper is of a good standard and a timely contribution to the literature on a useful topic. I therefore support the publication of this article, but recommend some minor revisions to the manuscript beforehand, summarised below.
Requested changes
-
The discussion of the choice of prior could be expanded on. For example, using a Gaussian centred around the estimate beforehand seems a fair choice, but using the estimator \hat{\theta}} gives the impression it is calculated from something.
-
On the same topic, I understand why the authors wish not to have to restrict the support for the prior, but would it be possible to obtain meaningful results with a prior with a large variance? What about if we have a flat prior, then what would be the optimum strategy? This is relevant, as we may sometimes not be able to begin with a good estimate of the angle and as such need a method that works for all values.
-
A final question about the choice of prior - what would happen if we chose a biased prior? How strongly would this affect the choice of the optimal state? This may be beyond what the authors wish to show in this paper, but is perhaps worthy of commenting on.
-
The family of states that the authors show are optimal is well explained and intuitive, so I wonder if it would be possible to show that this is indeed the case, i.e. show that squeezing does indeed occur for these states? It is then clear to see why the probe states work in the way that they are described to.
-
Could the authors explain the seemingly different behaviour of the pink line for |\alpha|^2 / E = 1 in Fig. 2? From visual inspection, it seems the curves are gradually tending towards the optimum for increasing |\alpha|^2 / E, but the change from 0.9 to 1 creates a sudden jump in the trend. Moreover, the choice of parameters should be clearly defined. For example, it is not clear what the choice of the squeezing parameter r is. In Fig. 3, it is stated that this is taken to be some optimum value, but it should be reinforced what this actually corresponds to.
-
The appendices are not well referred to within the text. For example, the main body references Fig. 6, which occurs in an appendix, but does not ever refer to the appendix directly. The authors should review the formatting to make sure these appendices are appropriately incorporated.
-
When the Fisher information is calculated in the Bayesian estimation protocol, the authors should state how this is done more clearly. For example, is the calculated Fisher information the result from the frequentist methods for fixed value(s) of the parameter, or is it obtained directly from the resulting posteriors?
Recommendation
Ask for minor revision

---

## Round 3 · Referee Report · Robert Trenyi (Referee 1) · 2025-4-17

Report

The authors have addressed all my questions in detail. I have no further questions and I recommend the publication of the manuscript in its current form.

Recommendation

Publish (meets expectations and criteria for this Journal)

---

## Round 3 · Referee Report · Anonymous (Referee 2) · 2025-4-28

Report

The authors have provided detailed responses to all of the points raised by the referees, and improved their manuscript accordingly. Based on this, I am happy to support the publication of this work.

Recommendation

Publish (meets expectations and criteria for this Journal)

---

## Round 3 · Author Response

We would like to thank both referees for their time to read and assess our article and for their useful and constructive comments and criticisms. Below we address their comments and questions point by point.

Regards, the authors

Reply to referee 1

We are glad to hear the positive evaluation of our manuscript.

R1: I have the following suggestions/questions, and also found some typos: At the end of the introduction there should be a brief sentence about the content of the appendices. Regarding the previous point I did not find direct references to Appendix B, C in the main text (apart from referring to Fig. 6 that is in Appendix C), even though it contains useful information.

A: We have added a sentence about the content of the appendices and also included references to the appendices throughout the manuscript.

R1: 6th line of the 2nd paragraph of III.A.: The vector r should contain (<q>,) instead of (<q>,<q>)? It would be useful to define the anticommutator {,} when defining the covariance matrix in the second paragraph of III.A. „In this section, we describe the estimation scenario that we are looking at.” sounds too informal to me (Section III. beginning) III.B: typo in title homodye -> homodyne In Eqs. (14) and (16) the entries of the matrices should be better spaced for readability, if possible, in this two-column format 1st reference: Americal Journal ->American Journal

A: We are grateful to the referee for spotting these typos and stylistic shortcomings, we have followed the referee’s suggestion.

R1: Can the authors give a simple intuitive argument already at the beginning of IV.C. why is it better to allocate all the energy in the squeezing of the state for the frequentist approach?

A: In the frequentist approach one tries to find the optimal estimation strategy locally around a given parameter. This is reflected by the maximization of the FI as the figure of merit. We consider the estimation of a rotation of a state in phase space with homodyne measurements in a given quadrature, which without loss of generality we assume to be the position q. For squeezed vacuum states the uncertainty ellipse in phase space is elongated in one quadrature and squeezed in the other. Among states of a given energy, the squeezed vacuum states have the smallest uncertainty in the appropriate quadrature. This makes them universally powerful probe states for local estimation, compare for example with Ref. [16]. The finding that they are indeed optimal also in the considered scenario and even locally saturate the Quantum Cramér-Rao bound was already known, see Ref. [15]. When considering global estimation, the high uncertainty in the complementary quadrature makes squeezed states quickly worse. This fact is also reflected in the peaked shape of the FI in Fig. 5. We have added this explanation in the manuscript.

R1: What is the intuitive reason that the parameter varphi changes significantly between the two approaches (observation of IV.C)?

A: Both in the local analysis and in the Bayesian analysis for sufficiently small variances of the prior, the optimal probe state is a squeezed vacuum state. In the local approach, the optimal angle of the probe is given by varphi=arccos(tanh{2r})-2hat{theta}_0. This angle is also approached in the Bayesian analysis when the variance of the prior goes to zero, see Fig. 6. That the optimal angle varphi in the Bayesian analysis changes quickly with increasing variance, might stem from the asymmetric shape of the peak of the FI in Fig. 5. When averaging over a larger region, it might be beneficial to include more weight from the slower decreasing right slope, than from the quickly decreasing left slope. This effect might also be increased by the fact that the secondary peak, even though having high FI, leads to a wrongful estimation of the phase and is thus avoided in the Bayesian analysis. We have clarified this point in the manuscript.

R1: Is there a particular reason why varphi=2pi-2theta is used instead of the previously used varphi=-2theta in the penultimate paragraph of Section IV.C?

A: Rotations are indistinguishable for angles differing by 2pi, so the two are identical. For consistency, we have unified the notation in the revised version of the manuscript.

R1: For Eq. (16), it should be stated which equation is evaluated from II.A for clarity.

A: Both Eq. (3) and (5) can be used to calculate the quantum Fisher information in this instance. We do not explicitly do this in the manuscript; Eq. (16) is taken from Ref. [15] and Ref. [32].

R1: Do I understand correctly that the authors claim that there cannot be other optimal strategies apart from HUS and LUS based on numerical evidence?

A: Within the family of pure single-mode Gaussian states and the range of energies and priors we considered, numerical optimization consistently indicated one of these two strategies to be optimal, independently of the starting point of the optimization. Given that the parameter space has only 3 dimensions, we trust that these are indeed the only optima. This is consistent with the fact that the LUS strategy converges to the locally optimal strategy saturating the quantum Cramer-Rao bound, and is backed up by the fact that there exists an explanation for why the HUS is optimal for larger uncertainties. Although it is likely that this result extends also to other values of energy and different priors, one has to be careful with such claims. In contrast, when extending the possible probe states to arbitrary states, it is not only possible, but most likely, that different families of states become optimal.

R1: What exactly is the difference between the “simplified” and the “pushed further” strategies in Section V? The authors should clarify this a bit more in my opinion.

A: In principle, the optimal probe state depends on the distribution encoding our knowledge of the parameter. Therefore, after each measurement, the optimal probe state has to be determined numerically. If we instead assume that the distribution is close to a Gaussian distribution, we can use the probe state that we know is optimal from the calculation done beforehand, simplifying the process. Instead of depending on the whole distribution the probe state that is used in the next round only depends on its mean and its variance. We can simplify this process one step further. Instead of making the probe state dependent on the variance of the distribution at a given point in the estimation, we let it depend on the expected variance at that point of the estimation. This allows to design in advance the probe states that are going to be used, up to a global phase that will still depend on the measurement outcomes. We agree that this detail was not explained clearly and have reformulated the corresponding paragraph.

R1: Would it be possible to obtain meaningful results for the case when the noise does not commute with the dynamics? Maybe the authors can comment on that.

A: When the noise does not commute with the dynamics, it is not so easy to draw analogies with our approach. In case the noise commutes with the estimated dynamics, one can interpret the noise as uncertainty in the parameter, and therefore, we can encode the noise in the prior distribution. In the case of non-commuting noise, this is no longer the case. It might be possible to split the noise into a commuting and a non-commuting part. The commuting part can then be modelled as uncertainty in the estimation. The non-commuting part is usually not as damaging to the estimation. This is an interesting question that might be worth studying in more detail.

R1: Just out of curiosity: Are there results for general non-Gaussian states? What can they achieve in each metrological setting?

A: In our work, we focused on Gaussian states, which are also widely used because they are comparably easy to prepare in the lab and robust against noise. Nevertheless, non-Gaussian NOON states are asymptotically optimal for phase estimation and widely used in the literature. For instance, in Ref. [26], the authors used NOON states in the non-asymptotic regime and showed they are optimal when measuring the quadratures.

R1: Also out of curiosity: Do similar results also hold for other types of measurements (like heterodyne)?

A: Generally, for any measurement that is not the eigenbasis of the symmetric logarithmic derivative, the Fisher information depends on the value of the estimated parameter. The optimal probe state will therefore generally depend on the knowledge of the estimated parameter. States that have a high, but peaked FI might be optimal for low uncertainties, whereas states with a lower but broader FI might be better for high uncertainties. Similar results should also be obtained for heterodyne measurements. Nevertheless, homodyne detection might be an extreme case. On one hand, as it is known in the literature, it can achieve Heisenberg scaling. On the other hand, a wrong phase of the squeezed vacuum probe state achieving Heisenberg scaling does not result in any information about the parameter.

We would like to thank the referee for their careful evaluation of our manuscript. We have followed their suggestions and hope that we were able to clarify the raised questions.

Reply to referee 2

We thank the referee again for their careful reading and are glad that our results are found interesting and useful.

R2: 1. The discussion of the choice of prior could be expanded on. For example, using a Gaussian centred around the estimate beforehand seems a fair choice, but using the estimator hat{theta}} gives the impression it is calculated from something.

A: The prior distribution encodes our knowledge or belief of the parameter before the estimation. The mean of this distribution is our estimate, that is used to design the probe state, even though it does not necessarily depend on any measurement. We agree that the choice hat{theta} to denote this might be confusing, we have changed it to hat{theta}_0 to highlight the difference to the estimator hat {theta}. In the repeated measurement scenario, the estimate is obtained from the posterior distribution of the previous round and depends on the outcome m obtained hat{theta}(m). We have clarified this point in the revised version of the manuscript.

R2: 2. On the same topic, I understand why the authors wish not to have to restrict the support for the prior, but would it be possible to obtain meaningful results with a prior with a large variance? What about if we have a flat prior, then what would be the optimum strategy? This is relevant, as we may sometimes not be able to begin with a good estimate of the angle and as such need a method that works for all values.

A: We restrict the support of the prior in this specific way because homodyne measurements do not give us any information on the complementary quadrature. That is, measuring in q gives us no information about p, so we can not distinguish between states mirrored by the q-axes. Moreover, in Ref. [29] the authors studied the flat prior on the interval [0,pi]. In principle, any family of distributions is possible, uniform step functions could also be a possible choice. To us, Gaussian distributions seem like a natural choice to encode our knowledge about the estimate and the variance. Additionally, the Bernstein–von Mises theorem states that after enough measurement rounds the posterior distribution converges towards a Gaussian distribution. In that sense, a Gaussian prior could encode data from previous measurement rounds. Gaussian priors are a natural choice, but certainly not the only meaningful one. We have clarified this point in the revised version of the manuscript.

R2: 3. A final question about the choice of prior - what would happen if we chose a biased prior? How strongly would this affect the choice of the optimal state? This may be beyond what the authors wish to show in this paper, but is perhaps worthy of commenting on.

A: If we understand it correctly, a biased prior is a function that is not symmetric around its mean. The choice of prior should reflect the knowledge about the parameter at a given stage of the estimation. If there is a good reason to assume that this distribution is not symmetric, a biased prior is a perfectly valid choice. It is likely that a different prior changes the optimal probe state and the estimation itself. However, we want to emphasize that the prior should not be seen as a part of the estimation process itself, that can be optimized, but merely as a function encoding our knowledge about the parameter. We have clarified this point in the revised version of the manuscript.

R2: 4. The family of states that the authors show are optimal is well explained and intuitive, so I wonder if it would be possible to show that this is indeed the case, i.e. show that squeezing does indeed occur for these states? It is then clear to see why the probe states work in the way that they are described to.

A: In this article, we show two families of optimal states that we identify as HUS and LUS. The LUS is optimal when the prior variance is small, the optimal states are squeezed vacuum states where all the energy is in the squeezing of the state. In the limit of small variances they converge towards the states that maximize the Fisher information, see Ref. [15]. The HUS is optimals when the prior variance is bigger, here the optimal states are squeezed and displaced states that have some fraction of the energy on the displacement and the rest on the squeezing. This fraction depends on the total energy and on the prior variance and is obtained through numerical analysis (see Fig. 1 for the numerical results). To sum up, the amount of squeezing that the optimal states have depends on the prior variance and the total energy of the probe.

R2: 5. Could the authors explain the seemingly different behaviour of the pink line for |alpha|^2 / E = 1 in Fig. 2? From visual inspection, it seems the curves are gradually tending towards the optimum for increasing |alpha|^2 / E, but the change from 0.9 to 1 creates a sudden jump in the trend. Moreover, the choice of parameters should be clearly defined. For example, it is not clear what the choice of the squeezing parameter r is. In Fig. 3, it is stated that this is taken to be some optimum value, but it should be reinforced what this actually corresponds to.

A: The optimum ratio of energy into displacement to total energy |\alpha|^2/E lies inside the interval [0.6-1], depending on the variance of the prior, see Fig. 1. For variance V_prior=0.2, the optimum is close to 0.9; therefore, the brown line |\alpha|^2/E=0.9 performs best. After the optimum, the trend is reversed, the two lines 1 (pink) and 0.8 (purple) perform similarly good. For smaller variances the purple line becomes better, whereas the pink line becomes worse. Regarding the choice of parameters, the energy of our pure Gaussian states E = |\alpha|^2 + \sinh^2{r} can be decomposed into a displacement part (depending only on |alpha|^2) and a squeezing part (depending only on r). For a fixed energy the choice of alpha determines the value of r=\arcsinh\sqrt{E-|alpha|^2}. We numerically calculate the optimal value of the displacement |\alpha|^2 that minimizes the average posterior variance. Fig. 3 compares the two strategies for different values of energy. The HUS shown as dotted lines has the optimal displacement |\alpha|^2 calculated before, compare to Fig. 1, and the optimal squeezing calculated as r=\arcsinh\sqrt{E-|alpha|^2}. These are numerical datasets, we do not have an analytical expression for the optimal parameters |alpha|^2 and r.

R2: 6. The appendices are not well referred to within the text. For example, the main body references Fig. 6, which occurs in an appendix, but does not ever refer to the appendix directly. The authors should review the formatting to make sure these appendices are appropriately incorporated.

A: We thank the referee for pointing this out, we have properly referred to the appendix and figures within in the revised version of the manuscript.

R2: 7. When the Fisher information is calculated in the Bayesian estimation protocol, the authors should state how this is done more clearly. For example, is the calculated Fisher information the result from the frequentist methods for fixed value(s) of the parameter, or is it obtained directly from the resulting posteriors?

A: The FI is always calculated using the formula derived in Appendix A, which will depend on the parameter theta that we want to estimate and the probe state. In the local scenario, one assumes a sufficiently good estimate of the parameter and can design the probe state in such a way that it maximizes the FI. When the knowledge of the parameter is encoded in a prior, one can find an upper bound on the estimation precision by averaging over the FI and the corresponding prior probabilities. The local optimal state maximizes the FI, the bound for Bayesian estimation depends also on FI for other values of the parameter.

We would like to thank the referee again for their time and constructive feedback. We have implemented the suggested changes and hope that with this the manuscript is now deemed appropriate for publication.

Docutils System Messages

---

## Round 3 · List of Changes

We have implemented the suggested changes, see the above reply. These changes include typos and small imprecisions, as well as some clarifications, mostly in section IV. and V.

---

## Editorial Decision

published